# Analysis of Prevalence and Predictive Factors of Long-Lasting Olfactory and Gustatory Dysfunction in COVID-19 Patients

**DOI:** 10.3390/life12081256

**Published:** 2022-08-17

**Authors:** María A. Callejón-Leblic, Daniel I. Martín-Jiménez, Ramón Moreno-Luna, Jose M. Palacios-Garcia, Marta Alvarez-Cendrero, Julissa A. Vizcarra-Melgar, Carlos Fernandez-Velez, Isabel M. Reyes-Tejero, Juan Maza-Solano, Jaime Gonzalez-Garcia, Beatriz Tena-García, María E. Acosta-Mosquera, Alfonso Del Cuvillo, Serafín Sánchez-Gómez

**Affiliations:** 1Rhinology Unit, Department of Otolaryngology, Head and Neck Surgery, Virgen Macarena University Hospital, 41009 Seville, Spain; 2Biomedical Engineering Group, University of Seville, 41092 Seville, Spain; 3Rhinology Unit, Virgen de Valme University Hospital, 41014 Seville, Spain; 4Rhinology and Asthma Unit, ENT Department, The University Hospital of Jerez, 11407 Jerez de la Frontera, Spain

**Keywords:** olfactory disorders, COVID-19, prediction model, anosmia, parosmia, smell disorders, smell test, UPSIT

## Abstract

Background: Although smell and taste disorders are highly prevalent symptoms of COVID-19 infection, the predictive factors leading to long-lasting chemosensory dysfunction are still poorly understood. Methods: 102 out of 421 (24.2%) mildly symptomatic COVID-19 patients completed a second questionnaire about the evolution of their symptoms one year after the infection using visual analog scales (VAS). A subgroup of 69 patients also underwent psychophysical evaluation of olfactory function through UPSIT. Results: The prevalence of chemosensory dysfunction decreased from 82.4% to 45.1% after 12 months, with 46.1% of patients reporting a complete recovery. Patients older than 40 years (OR = 0.20; 95% CI: [0.07, 0.56]) and with a duration of loss of smell longer than four weeks saw a lower odds ratio for recovery (OR = 0.27; 95% CI: [0.10, 0.76]). In addition, 28 patients (35.9%) reported suffering from parosmia, which was associated with moderate to severe taste dysfunction at the baseline (OR = 7.80; 95% CI: [1.70, 35.8]). Among the 69 subjects who underwent the UPSIT, 57 (82.6%) presented some degree of smell dysfunction, showing a moderate correlation with self-reported VAS (*r* = −0.36, *p* = 0.0027). Conclusion: A clinically relevant number of subjects reported persistent chemosensory dysfunction and parosmia one year after COVID-19 infection, with a moderate correlation with psychophysical olfactory tests.

## 1. Introduction

Smell and taste dysfunction, symptoms frequently reported in upper respiratory tract infections caused by different types of viruses, became crucial during the COVID-19 outbreak due to their role as accurate predictors of the disease and their impact on quality of life [1,2,3,4,5,6]. Although some groups have made considerable research efforts to elucidate the causes of smell and taste dysfunction in COVID-19 [7,8], the underlying mechanism by which SARS-CoV-2 infection leads to chemosensory dysfunction remains unclear.

Although studies published to date have shown that the majority of patients recover their olfactory and gustatory function a few weeks after the COVID-19 infection, some patients suffer from persistent smell and taste loss beyond six months [1,9,10]. In fact, some authors have suggested that the chemosensory dysfunction caused by SARS-CoV-2 can be more severe than that caused by other seasonal viruses [11,12], and can frequently appear along with other qualitative disorders such as parosmia (i.e., misperception of an odor). Parosmia is one of the most relevant types of olfactory disorders, with a prevalence of 3.9% among the adult population [13], and has also been recognized as a long-lasting symptom of COVID-19, even in cases where the olfactory function is fully recovered [5,14,15]. Further studies are needed in order to clarify the actual prevalence and predictive factors leading to long-lasting chemosensory dysfunction and/or parosmia in COVID-19 patients [9,16,17].

In a previous work [6], the authors used statistical analysis and machine learning techniques to study the prevalence and predictive value of loss of smell and taste in a group of 777 patients suspected of COVID-19 infection. The results showed that loss of smell and taste were the most predictive symptoms, with an accuracy of up to 80% when using self-reported visual analog scales (VAS) as predictor variables. In this second study, we follow the evolution of COVID-19 symptoms in a group of confirmed positive patients who agreed to participate in a second visit one year after the primary infection. The aim of this study is to continue to analyze the long-term olfactory and gustatory impairment in COVID-19 patients using both self-reported VAS scores and psychophysical olfactory evaluation. This analysis should allow us to gain further knowledge of the evolution and recovery of chemosensory dysfunction during and after COVID-19 infection.

## 2. Materials and Methods

### 2.1. Study Design, Setting, and Participants

Figure 1 shows the workflow chart for this study. This is a prospective cohort study of subjects aged 18 or over who were diagnosed with COVID-19 and were assessed in four different hospitals in the south of Spain between March and April 2020. The COVID-19 infection was confirmed via RT-PCR testing of nasopharyngeal and oropharyngeal tract specimens. All subjects completed an initial questionnaire regarding the presence and severity of their symptoms, including loss of smell and taste. These patients were subsequently observed between March and April 2021 to analyze the evolution of their symptoms one year after the primary COVID-19 infection. Subjects were contacted via telephone or mail a maximum of three times in case of no response over a period of fifteen days. Patients who had a second positive diagnosis of COVID-19 during the follow-up period were excluded from the study. This study complies with the Declaration of Helsinki and was approved by the Ethics Committee of the Virgen Macarena University Hospital in Seville, Spain (Code Protocol: 1963-N-20), and all patients gave their informed consent to participate.

### 2.2. Study Variables

Demographic and clinical variables such as age and gender were collected, together with comorbidities such as diabetes, asthma, rhinitis, sinusitis, or allergy, among others. Self-reported scores quantifying the severity of symptoms during the COVID-19 infection, such as loss of smell, loss of taste, nasal obstruction, nasal discharge, facial pain, cough, and dyspnea, were reported using numerical visual analog scales (VAS) according to guidelines EPOS 2020 [18,19]. Through VAS, patients reported the severity of their symptoms by indicating a position along a continuous line ranging from 0 to 100. The presence of symptoms such as fever and diarrhea was also collected as binary variables. Fever was defined as a body temperature greater than 38 °C measured under the armpit either by clinicians or the patients themselves with a calibrated thermometer.

The same questionnaire was administered to volunteers who agreed to report the evolution of their symptoms one year after the COVID-19 infection. Patients were also asked to inform about the time of recovery of the olfactory and gustatory losses (in weeks) and the presence of olfactory disorders such as parosmia, by answering the following question: “Would you say that things smell either different or unpleasant to you now compared to before the infection?”, which was reported as a binary variable. A subgroup of patients also agreed to undergo a psychophysical evaluation of olfactory function through the Spanish-American version of the 40-odorant University of Pennsylvania Smell Identification Test (UPSIT) (Sensonics International, Haddon Hts., NJ, USA). The UPSIT is a standardized and widely used screening olfaction test with high internal reliability (test-retest, *r* = 0.94) [20]. Numerical UPSIT scores ranging from 1 to 40 were collected. Categorical diagnostic outcomes (normosmia, mild microsmia, moderate microsmia, severe microsmia, and anosmia) were also derived according to overall UPSIT score and gender [21].

### 2.3. Data Analysis and Statistical Methods

Categorical variables were reported as percentages, while continuous variables were reported as mean and standard error (SE) or median and interquartile range (IQR) values, depending on distribution. Differences in symptom prevalence at the baseline and the 12-month follow-up were evaluated through the McNemar test, while differences in mean VAS values were estimated through the paired *t*-test. The risk of suffering from persistent chemosensory dysfunction and/or parosmia one year after COVID-19 infection was estimated as an odds ratio (OR) through crude binomial logistic regression. Multiple logistic stepwise regression models (backward and forward) were calculated to adjust ORs. In addition, multiple linear stepwise regression models were calculated following the Bayesian Information Criterion (BIC) to predict the change in VAS scores at the baseline and one year after the infection (VAS [baseline]–VAS [12 months]). Final model results were reported using adjusted effect estimates (β), SE, 95% confidence intervals (CI), and *p*-values. The coefficient of multiple determination values (R^2^) was calculated to assess model fitting. Finally, differences in median UPSIT scores between unpaired groups were evaluated using the Kruskal–Wallis and post-hoc Dunn–Sidak tests. Correlation between VAS and UPSIT scores was evaluated through Pearson’s correlation coefficient. Data analysis was conducted using the IBM SPSS 28 statistical package and the Statistics and Machine Learning Toolbox in Matlab (v. R2022a, The MathWorks Inc., Natick, MA, USA). A *p*-value lower than 0.05 was considered significant and is reported in bold in our results below.

## 3. Results

Among the 421 patients who tested positive for COVID-19 in our first study [6], 102 patients agreed to complete a second questionnaire about the evolution of their symptoms 12 months after the primary COVID-19 infection (24.2%). The sociodemographic and clinical characteristics of the 102 patients are listed in Table 1. The cohort included 70 women (68.6%) and 32 men (31.4%), with a mean age of 46.8 ± 13.9 (range 23–89). The cohort was mainly composed of healthy adults with no comorbidities as follows: only 13 (12.8%, 95% CI: [7.0, 20.8]) reported rhinitis (*n* = 8, 61.5%), diabetes (*n* = 2, 15.4%), asthma (*n* = 1, 7.7%), sinusitis (*n* = 1, 7.7%) and Sjogren’s syndrome (*n* = 1, 7.7%).

### 3.1. Evolution of COVID-19 Symptoms

Among these 102 patients, the most frequently reported symptoms during COVID-19 infection were chemosensory dysfunction by 84 patients (82.4%, 95% CI: [73.6, 89.2]), cough by 76 patients (74.5%, 95% CI: [64.9, 82.6]) and nasal discharge by 59 patients (57.8%, 95% CI: [47.4, 67.6]). There were significant differences in the prevalence of all symptoms at the baseline and at the 12-month follow-up according to the McNemar test (Table 1). Additionally, mean VAS scores for all symptoms were significantly lower at follow-up than at baseline (see Table 2).

At 12 months, the most reported long-lasting symptoms were chemosensory dysfunction by 46 patients (45.1%, 95% CI: [35.2, 55.3]), nasal discharge by 35 (34.3%, 95% CI: [25.2, 44.4]), and dyspnea by 26 (25.5%, 95% CI: [17.4, 35.1]). Among the 84 patients who reported chemosensory dysfunction at the baseline, 39 (46.4%) also complained of nasal congestion. A similar percentage of patients (41.3%) reported these two symptoms at the 12-month follow-up.

### 3.2. Prevalence, Severity, and Time of Recovery of Chemosensory Dysfunction

The prevalence of chemosensory dysfunction decreased from 82.4% (95% CI: [73.6, 89.2]) to 45.1% (95% CI: [35.2, 55.3]) at the 12-month follow-up (*p* < 0.001). Both at the baseline (68.6%, 95% CI: [58.7, 77.5]) and 12 months later (26.5%, 95% CI: [18.2, 36.1]), the most common form of presentation was a combined chemosensory dysfunction affecting both smell and taste (Table 1).

Regarding the intensity of chemosensory dysfunction, the mean VAS for loss of smell decreased from 67.6 ± 42.3 at the baseline to 15.2 ± 25.9 (*t* = 12.788; *p* < 0.001) at 12 months (Table 2). During acute COVID-19 infection, 24 out of 102 patients did not suffer from a loss of smell (23.5%, 95% CI: [15.7, 33.0]). The intensity of the symptoms reported was mild (VAS > 0–30) in 3 out of 102 patients (2.9%, 95% CI: [0.6, 8.4]), moderate (VAS > 30–70) in 11 (10.8%, 95% CI: [5.5, 18.5]), and severe (VAS > 70–100) in 64 (62.7%, 95% CI: [56.2, 72.1]) during COVID-19 infection (Table 3). After one year, 59 out of 102 patients did not report a loss of smell (i.e., VAS = 0) (57.8%, 95% CI: [47.7, 67.6]). Specifically, 36 out of 78 patients who complained of a smell disturbance during the acute infection showed a total recovery, while 31 patients with moderate to severe loss of smell still reported persistent dysfunction but with decreased intensity. However, 3 patients did not experience any change, and 8 patients still reported a severe loss of smell one year after the infection (Table 3).

For loss of taste, the mean VAS score decreased from 59.7 ± 40.0 at the baseline to a value of 9.6 ± 20.7 (*t* = 11.724; *p* < 0.001) at the 12-month follow-up (Table 2) with 71 out of 102 patients (69.6%, 95% CI: [59.7, 78.3]) reporting no taste dysfunction (see Table 3). Among the 31 patients who did not completely recover their gustatory function, 25 did report an improvement in their perception of taste. Three patients remained with moderate or severe taste dysfunction, and one reported that their condition had worsened from moderate to severe at the 12-month follow-up. A strong correlation was found between the VAS reported for loss of smell and taste both at the baseline (Pearson’s *r* = 0.65, *p* < 0.01) and one year after the infection (Pearson’s *r* = 0.67, *p* < 0.01) in the study cohort.

The number of patients who recovered smell and taste before 2, 4, 24, and 48 weeks are shown in Table 4. Eighteen and 27 patients recovered smell and taste, respectively, before two weeks, and 8 and 9 before four weeks. Ten patients recovered smell and 11 patients recovered taste before 24 weeks. No patient recovered smell or taste between 24 and 48 weeks. Finally, three patients who had not complained about chemosensory dysfunction at baseline respectively reported a mild loss of smell, a moderate loss of taste, and a severe loss of taste 12 months after infection.

In the regression analysis, the persistence of chemosensory dysfunction at 12 months was neither associated with the VAS scores reported for the severity of loss of smell and taste during the acute COVID-19 infection nor with those reported for other symptoms such as nasal congestion and discharge, facial pain, cough, or dyspnea. However, having suffered from fever at the baseline reduced the risk of persistent chemosensory dysfunction at 12 months (OR = 3.19; 95% CI: [1.28, 7.94], *p* = 0.0126) in the crude analysis (Table 5). After adjusting for covariates, patients older than 40 years showed a lower odds ratio (OR) for recovery (OR = 0.20; 95% CI: [0.07, 0.56], *p* = 0.0023). In addition, a duration of the loss of smell in excess of four weeks saw a decrease in the probability of recovery (OR = 0.27; 95% CI: [0.10, 0.76], *p* = 0.0133).

Regarding the change or improvement in the intensity of loss of smell at the 12-month follow-up, a multiple linear regression model (Table 6) showed that this was significantly associated with age (β = −0.49; 95% CI: [−0.93,−0.04], *p* = 0.0327), smaller duration of loss of smell (β = −0.66; 95% CI: [−1.10, −0.21], *p* = < 0.0001), and higher VAS scores at the baseline (β = 0.86; 95% CI: [0.68, 1.04], *p* = < 0.0001). Regarding loss of taste, this was also showed to be associated with duration (β = −0.79; 95% CI: [−1.17, −0.42], *p* < 0.0001), and VAS score at the baseline (β = 1.07; 95% CI: [0.93, 1.21], *p* < 0.0001).

### 3.3. Prevalence of Parosmia after COVID-19 Infection

Twenty-eight patients reported suffering parosmia 12 months after the COVID-19 infection. The distribution of parosmia in relation to the intensity of the olfactory loss both at the baseline and at the 12-month follow-up is presented in Table 7. It should be noted that 24 out of the 28 patients (85.7%, 95% CI: [67.3, 96.0]) that reported parosmia at 12 months had previously complained of a severe loss of smell during the primary infection. Among these 24 patients, 2 had completely recovered their olfactory function, 12 and 4 patients remained with mild and moderate symptoms, and 6 had persistent severe dysfunction at the 12-month follow-up.

Parosmia was associated with moderate and severe intensities (VAS > 30) of smell dysfunction at the baseline (OR = 6.27; 95% CI: [1.32, 29.87], *p* = 0.0211) when adjusted for significant covariates (Table 8). Additionally, a duration of loss of taste longer than 12 weeks was associated with a higher risk of parosmia one year after the infection (OR = 6.16; 95% CI: [1.74, 21.86], *p* = 0.0049).

### 3.4. Psychophysical Evaluation of Olfactory Function

Among the 102 patients who agreed to complete the second questionnaire 12 months after the COVID-19 infection, 69 (67.7%) also volunteered to undergo psychophysical evaluation of olfactory function through the UPSIT test. Out of these 69 patients, 48 (69.6%) were female and the mean age was 47.9 ± 13.9 years (range: 27–78 years), with only 5 patients being older than 65 years. There were no significant differences in median age (*p* = 0.27), gender (*p* = 0.77), or VAS score at the baseline for loss of smell (*p* = 0.18) between subjects who underwent UPSIT and subjects who did not.

The median UPSIT score was 31.0 (IQR = 5.0). Among the 69 subjects who underwent the test, 57 (82.6%) showed some degree of smell dysfunction. Thirty patients (43.5%, 95% CI: [31.6, 56.0]) had mild microsmia, 15 (21.7%, 95% CI: [12.7, 33.3]) moderate microsmia, 7 (10.1%, 95% CI: [4.2, 19.8]) severe microsmia, and 5 patients (7.2%, 95% CI: [2.4, 16.1]) had anosmia (Table 9). Among the 32 patients who self-reported normal sense of smell at the 12-month follow-up, UPSIT exhibited normal smell or normosmia in only 8 (25.0%), while mild microsmia was detected in 13 (40.6%), moderate microsmia in 8 (25.0%), severe microsmia in 2 (6.3%), and anosmia in 1 (3.1%). A Kruskal–Wallis test detected differences in UPSIT scores between groups (χ^2^ = 10, *p* = 0.0153), with post-hoc Dunn–Sidak revealing significantly higher median UPSIT scores of 32 (IQR = 5) in patients reporting no loss of smell, compared with that of 24 (IQR = 10) in those reporting a severe loss of smell (*p* = 0.0121) (Figure 2a). The correlation between UPSIT scores and self-reported VAS for loss of smell 12 months after the COVID-19 infection was moderate (Pearson’s *r* = −0.3562, *p* = 0.0027) (Figure 2b), although higher among women (Pearson’s *r* = −0.5164, *p* < 0.001). Additionally, the correlation of UPSIT scores with age was also moderate (Pearson’s *r* = −0.3486, *p* = 0.0033). A multiple regression analysis confirmed the association of UPSIT scores with age and self-reported VAS scores 12 months after the COVID-19 infection (Table 10).

## 4. Discussion

The evolution of smell and taste disorders in COVID-19 patients continues to be an ongoing topic of study. This prospective study includes mild-to-moderate positive COVID-19 patients infected with the first variant of SARS-CoV-2 between March and April 2020 in the south of Spain, with the aim of analyzing the evolution of smell and taste disorders, as well as other common symptoms, one year after the primary infection. The main outcomes of our study are the following: (1) chemosensory dysfunction is highly prevalent during the acute phase of COVID-19 infection, (2) VAS for smell and taste improve at the 12-month follow-up; however, approximately half of patients continue to suffer from some degree of dysfunction with varying recovery times, (3) parosmia is a long-lasting symptom after COVID-19 infection, although a lower frequency than that reported in previous studies was found in our study, and (4) psychophysical evaluation of olfactory function through UPSIT shows a moderate correlation with self-reported VAS measures for loss of smell one year after COVID-19 infection.

### 4.1. Prevalence of Chemosensory Dysfunction in COVID-19 Subjects

Although only 24.2% of the selected patients volunteered to participate in the second phase of the study, a sample of 102 patients with similar demographic characteristics to those in previous studies (mean age: 46.8 ± 13.9 years, 68.6% women) [16,22,23,24,25,26] was achieved. As expected, a significantly lower prevalence and severity were reported at the 12-month follow-up for all symptoms analyzed (Table 1 and Table 2). Chemosensory dysfunction, understood as a loss of smell or taste, was the most prevalent symptom during the infection in our sample (82.4%, 95% CI: [73.6, 89.2]), similar to that reported in [9,17,27,28,29,30,31,32]. More than half of the subjects (68.6%, 95% CI: [58.7, 77.5]) had a combined alteration of both smell and taste, which is within the range of 34% to 86.7% reported in previous studies [24,33,34,35]. An isolated smell or taste disorder appeared at considerably lower frequencies, 7.8 and 5.9%, respectively, during the infection, which is close to the 10% reported in [29,34]. Regarding the intensity of symptoms, several authors have highlighted the high prevalence of patients with severe loss of smell and taste during infection, with reported frequencies between 51.8% and 86.4% for smell, and from 45.5% to 78.9% for taste [9,27,29,31,33,36]. These previous results coincide with those in our study, where severe losses of smell and taste were observed in 62.7% and 51.0% of patients, respectively.

### 4.2. Evolution, Severity, and Time of Recovery of Chemosensory Dysfunction

At the 12-month follow-up, chemosensory dysfunction was still reported in 45.1% of patients. The smell function was fully recovered in half of the patients (57.8%), and one-third (30.4%) reported a subjective improvement in VAS, whereas 10.8% of patients did not report any improvement. In the same way, 69.6% of the patients reported a full recovery of taste at 12 months, and 24.5% saw some improvement in their symptoms (Table 3). Concretely, the mean VAS improvement for smell was 52.4 ± 41.4 (t = 12.8; *p* < 0.001) and for taste, 50.2 ± 43.2 (*t* = 11.7; *p* < 0.001) (Table 2). This is consistent with a previous analysis that showed an improvement of around 50 points six and seven months after the infection [32,37]. However, it should be noted that high rates of chemosensory dysfunction were still found in our study during follow-up, even after improvement of the subjective VAS scores. Although the frequency of chemosensory dysfunction reported in the follow-up varies considerably in the literature from 16.8 to 54.5% [17,26,33,38,39], a rate similar to that obtained in our study (near 50%) has been reported in several previous studies [9,26,32,37,39,40]. Such variability might be explained not only by the differences in the patients’ characteristics and follow-up periods but also by the inherent limitations of the subjective self-assessment of olfactory function [2].

The time of recovery for chemosensory dysfunction after COVID-19 infection is described in Table 4. Approximately half of the patients who recovered smell and taste reported a full recovery within the first two weeks, in line with the early recovery times seen in other studies [8,22,38,41]. However, the other 50% of patients who recovered their sense of smell did so between 4 and 24 weeks [26,40]. This issue has increasingly gained the attention of researchers who have tried to elucidate the predictor factors accounting for persistent chemosensory dysfunction in the long term [22,23,42]. The role demographic factors such as age, gender, or comorbidities play in the prognosis of chemosensory recovery remains uncertain. While some authors have found earlier recovery times in younger subjects [42,43], others have not found age to be a relevant factor [41]. In our study, a multiple logistic regression model associated higher ages (OR = 0.20; 95% CI: [0.07, 0.56], *p* = 0.002) and a duration of loss of smell longer than four weeks (OR = 0.27; 95% CI: [0.10, 0.76], *p* = 0.013) with a higher risk of persistent dysfunction. We did not find any association between the presence of comorbidities and the persistence of a chemosensory impairment, in concordance with previous studies, where the prevalence of comorbidities was even higher than that of 12.8% found in our study [23,28,29,44]. In addition, the analysis of the effects of other nasal and systemic symptoms during the infection has led to divergent results in different studies. Whereas some authors relate these symptoms to worsened results in the recovery of smell and taste both in the short and long term [26,39,43,44], others have not found any such relation [31,35,38]. In our study, we have shown that suffering from fever at the baseline was independently associated with a higher odds ratio for chemosensory recovery (OR = 3.19; 95% CI: [1.28, 7.94], *p* = 0.0126). However, no symptom was significantly associated with a higher risk of persistent chemosensory dysfunction in the multivariable analysis, not even nasal congestion, which had a larger prevalence of 40% in our sample compared with lower rates of 12% and 30% reported in [9,16,24,33]. Therefore, our results are consistent with previous works that highlight the lack of clinical factors associated with smell and taste disturbances in COVID-19 infection [31,38].

Non-statistically significant associations were found between the severity of smell and taste symptoms at the baseline and a higher risk of persistent chemosensory dysfunction at the 12-month follow-up (Table 5). Indeed, patients who reported higher VAS scores at the baseline showed on average a greater change in VAS scores, according to the respective analysis of loss of smell and taste through a multiple linear stepwise regression model (Table 6). Age and longer time periods for recovery were associated with lower changes in VAS scores for both smell and taste [26,33,45].

### 4.3. Prevalence of Parosmia 12 Months after COVID-19 Infection

The prevalence of parosmia at the 12-month follow-up was 27.4% in our study, similar to that reported in [38], but lower than the 43.1% reported at the 6-month follow-up in [9] and the 46.8% reported 11 months after the infection [39]. Other studies with follow-up times ranging from seven to eighteen months have reported a rate of parosmia above 60% [15,46,47]. Late-onset times for parosmia have also been described even in patients who did not perceive any smell dysfunction during the acute infection [11,48,49]. Similarly to [9], most of the patients who suffered parosmia in our study (*n* = 24) self-reported a severe loss of smell during the acute infection, with 21.4% reporting the same severity at the 12-month follow-up (Table 7). Indeed, according to simple logistic regression models, moderate or severe combined smell and taste dysfunction and a duration of loss of smell and taste longer than three months were associated with a higher risk of parosmia at the 12-month follow-up. Conversely, having developed fever during the acute infection resulted in being a protective factor for persistent parosmia (OR = 0.40; 95% CI: [0.16, 0.96], *p* = 0.0407). In [23], other symptoms, such as cough and dyspnea, were found to act as protective factors for parosmia. Although no direct relationship between fever and the risk of suffering from long-lasting parosmia had previously been detected, Tham et al. reported a negative association between these two symptoms during the acute phase of the disease [50]. However, in a similar way to the results obtained for chemosensory recovery in Table 5, this significant effect was no longer seen in the multivariable analysis, highlighting the fact that further studies are still needed in order to corroborate the protective effect observed in our study.

In line with the literature, variables such as the severity of olfactory loss during the acute infection (OR = 6.27; 95% CI 1.32–29.87, *p* = 0.021) [41,51] and a duration of the loss of taste longer than 12 months (OR = 6.16; 95% CI 1.74–21.86, *p* = 0.005) [33] had a greater impact on the multivariable analysis of the risk of developing parosmia (Table 8). Furthermore, although a strong correlation (*r* > 0.60) was found between VAS scores for smell and taste both at the baseline and at the 12-month follow-up, our results in the multivariable analysis suggest that in order to better prognosticate long-lasting disorders such as parosmia, smell and taste have a different effect and should be analyzed separately. A comprehensive analysis of both senses could help better understand the reported influence of olfaction in the perception of taste as well as the influence of an altered taste in the anomalous perception of odors [52,53,54].

### 4.4. Psychophysical (UPSIT) Evaluation of Olfactory Function 12 Months after COVID-19 Infection

Finally, although the assessment of olfactory function through subjective olfactometry testing is not widely used in clinical practice, it has increased considerably during and after the COVID-19 pandemic [9]. In our study, an UPSIT test was performed in a non-randomized subgroup of 69 volunteer patients who shared homogeneous characteristics and was therefore representative of the overall sample. The majority of patients (82.6%) showed an altered smell function in the UPSIT test, a considerably higher proportion than the 41.1% found in [55]. Other groups that quantified the smell through different olfactometry tests, such as the Sniffin’ Sticks test and the Brief Smell Identification test, showed rates for hyposmia from 25.4% to 72.5% [23,56] and 76% [57], respectively, compared with the 75.3% obtained in our study. The prevalence of anosmia in our study (7.2%) (Table 9) was also higher than rates previously published between 0% and 4% [55,56]. Furthermore, it is noteworthy that among the patients who self-reported a normal sense of smell, 75% exhibited some degree of alteration in UPSIT. A well-known inverse correlation between UPSIT scores and age [56] was found in our study (*r* = −0.35, *p* = 0.0033). As expected, although higher UPSIT scores were found in patients who reported lower smell VAS values, the correlation between UPSIT and VAS scores was only moderate (*r* = −0.36, *p* = 0.003) (Figure 2), which agrees with previous olfactometry studies conducted in the European population [10,26,56]. Although the UPSIT is a reliable, validated, and standardized smell test [58], it only evaluates odors qualitatively, which has been described as its major drawback [59]. As such, further validation studies in the Spanish population would be needed in order to account for the uncertainty and the moderate correlation values reported [26].

### 4.5. Limitations

One of the limitations of this study is the selection bias caused by the loss of patients in the follow-up, with the participation of only 24.2% of volunteers one year after infection. This is a well-known source of bias in clinical studies, especially when the sample is formed by a majority of healthy subjects who are assessed during long follow-up periods [60]. In addition, a recall bias might also have influenced the accuracy of the responses given by the patients one year after infection. Although our results agree with previous studies that included similar numbers of patients and varying follow-up periods (from 2 to 18 months) [9,16,17,26,32,39,40,51], we acknowledge that these two sources of bias may have led to an overestimation of the overall prevalence and prognostic factors found in this study.

Another limitation of our study may be due to the appearance or worsening of chemosensory dysfunction caused by other (non-controlled) upper respiratory tract infections during the follow-up period. Although this confusion factor could have led to a selection bias, we should highlight that the protection and isolation measures used for respiratory tract infections during the pandemic helped prevent infections from all respiratory viruses [57,61], and consequently, we can claim that the long-lasting chemosensory impairment seen in our study at the 12-month follow-up may be mostly attributed to the primary SARS-CoV-2 infection.

On the other hand, we emphasize that this study was developed during the outbreak of the first wave of COVID-19, which limits the results and conclusions derived to the first variant of the SARS-CoV-2 virus. In this sense, there is heterogeneity in the groups of patients studied in the literature, which cover different pandemic periods including the Gamma, Delta, and Omicron waves [62]. Prevalence studies over the different pandemic periods are needed in order to better quantify and compare the actual prevalence of chemosensory disorders caused by different SARS-CoV-2 variants, as well as to better understand and assess the evolution and long-term effects of COVID-19 infection in smell and taste disorders.

In this sense, the lack of consensus in the definition and analysis of smell and taste disorders and the heterogeneity of the results published so far make it difficult for a comparison to be made. Different terms such as chemosensory dysfunction, loss of smell, and loss of taste have been used interchangeably in the literature without comprehensively addressing the characteristics of the dysfunction (for instance, if this was combined or isolated). In addition, many authors have only focused on smell dysfunction without evaluating taste. We believe that a consensus on the analysis and reporting of smell and taste disorders should be reached by different research groups. Indeed, according to our results, we suggest that smell and taste should be analyzed and reported separately [52,53,54]. This could help validate and corroborate basic research studies that have proposed different pathophysiological mechanisms to explain the causes of loss of smell and taste separately, such as the neurodegenerative involvement of the olfactory bulb and the brain [8], or the role of the facial, glossopharyngeal, and vagus nerves in the oral mucosa [36], respectively. Further studies are required in order to clarify the underlying phenomenon of long-lasting loss of smell and taste, as well as parosmia, in SARS-CoV-2 infection.

Finally, we would like to remark that a new paradigm has been promoted in the monitoring and treatment of patients with olfactory dysfunction after the pandemic. ENT specialists now face new challenges in order to systematically assess and treat long-lasting COVID-19 smell and taste disorders in clinical practice. The implementation of already-known psychophysical olfactory tests and the design of novel objective and subjective measures can help us elucidate the true causes of smell and taste disorders in SARS-CoV-2 infection as well as the prognostic factors leading to persistent disorders such as parosmia.

## 5. Conclusions

COVID-19 patients over the age of 40 and those who suffered from a loss of smell during a period greater than four weeks had an increased risk of persistent chemosensory dysfunction one year after the infection. In addition to the quantitative chemosensory disorder, qualitative disorders such as parosmia were associated with a poor self-reported quality of smell during the acute infection as well as a longer duration of taste dysfunction. Psychophysical olfactory evaluation through UPSIT showed a moderate correlation with self-reported VAS scores, which quantified the severity of long-lasting COVID-19 smell dysfunction at the 12-month follow-up. Despite there being an increased number of research studies in the field, more comprehensive studies with larger sample sizes and longer follow-up periods should be conducted in order to disclose new knowledge into the prognostic and predictive factors of persistent smell and taste disorders in COVID-19 infection.

## Figures and Tables

**Figure 1 life-12-01256-f001:**
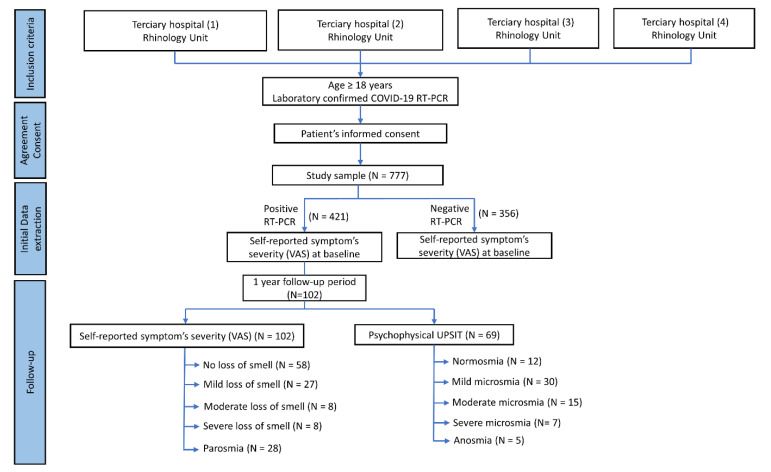
Workflow chart of our study including 777 subjects suspected of being infected by COVID-19 from March to April 2020 in four different hospitals from the south of Spain. All patients completed a baseline questionnaire reporting the severity of their symptoms, including loss of smell and taste, measured through quantitative visual analog scales (VAS) during the acute phase of COVID-19 infection. To confirm the COVID-19 diagnosis, all patients were subjected to RT-PCR. Patients with a confirmed diagnosis of COVID-19 were recruited one year later to study the evolution of their symptoms, especially loss of smell and taste. A second questionnaire based on self-reported VAS and psychophysical evaluation of olfactory function through the UPSIT test were conducted in a group of volunteers with N = 102 and N = 64 subjects, respectively. Differences between the severity of smell and taste symptoms were compared at the baseline and at the 12 month-follow-up, and the correlation with psychophysical UPSIT scores was evaluated. The prevalence and clinical factors leading to other self-reported qualitative disorders such as parosmia were also analyzed in our study.

**Figure 2 life-12-01256-f002:**
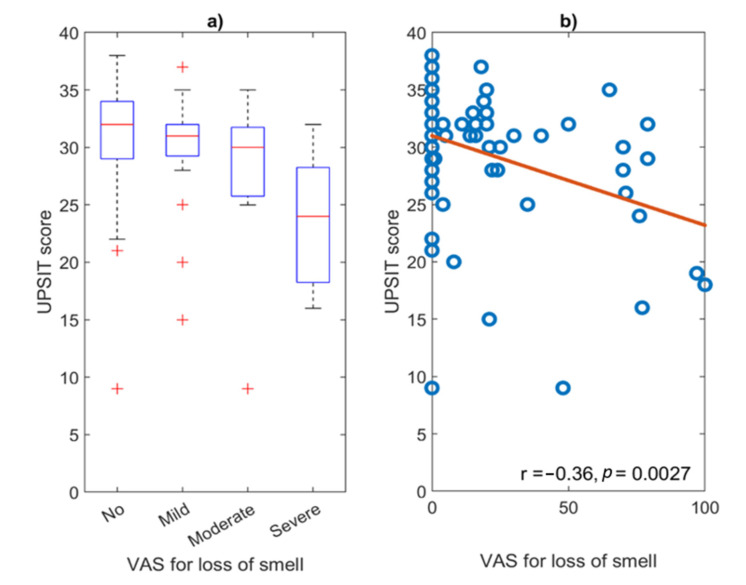
UPSIT scores at the 12-month follow up. (**a**) Boxplot distribution of UPSIT scores for different patient subgroups according to the severity of smell symptoms measured through VAS (No; mild; moderate, and severe) (**b**) Point distribution and correlation of UPSIT scores in relation to numeric VAS for loss of smell at the 12-month follow-up.

**Table 1 life-12-01256-t001:** Sociodemographic characteristics (age and gender), comorbidity, and prevalence of symptoms for 102 positive COVID-19 patients during acute infection and 12 months after the COVID-19 infection.

**Mean age (years) ± SD (Range)**	46.8 ± 13.9 (23–89)							
	** *n* **	**%**	**[95% CI]**					
**Gender**								
Male	32	31.4	[22.5, 41.3]					
Female	70	68.6	[58.7, 77.5]					
**Comorbidity**	13	12.7	[7.0, 20.8]					
**Symptoms**	**During acute COVID-19 infection**			**12 months post-COVID-19 infection**			**McNemar test**	
	** *n* **	**%**	**[95% CI]**	** *n* **	**%**	**[95% CI]**	** *χ* ** ** ^2^ **	** *p* **
**Chemosensory dysfunction**	84	82.4	[73.6, 89.2]	46	45.1	[35.2, 55.3]	<0.001	**<0.001**
Combined smell and taste dysfunction	70	68.6	[58.7, 77.5]	27	26.5	[18.2, 36.1]	0.0018	**<0.001**
Isolated smell dysfunction	8	7.8	[3.4, 14.9]	16	15.7	[9.2, 24.2]	0.0054	0.0768
Isolated taste dysfunction	6	5.9	[2.2, 12.4]	3	2.9	[0.6, 8.4]	0.0403	0.4531
**Nasal obstruction**	43	42.2	[32.4, 52.3]	27	26.5	[18.2, 36.1]	0.0358	**0.0149**
**Nasal discharge**	59	57.8	[47.7, 67.6]	35	34.3	[25.2, 44.4]	0.4586	**0.0011**
**Facial pain**	33	32.4	[23.4, 42.3]	14	13.7	[7.7, 22.0]	0.1286	**0.0017**
**Cough**	76	74.5	[64.9, 82.6]	17	16.7	[10.0, 25.3]	0.6844	**<0.001**
**Dyspnea**	56	54.9	[44.7, 64.8]	26	25.5	[17.4, 35.1]	0.4308	**<0.001**
**Fever**	57	55.9	[45.7, 65.7]	2	2.0	[0.2, 6.9]	0.1079	**<0.001**
**Diarrhea**	57	55.9	[45.7, 65.7]	2	2.0	[0.2, 6.9]	0.2044	**<0.001**

CI: Confidence interval.

**Table 2 life-12-01256-t002:** Mean VAS scores for symptoms during acute infection and 12 months after the COVID-19 infection.

Symptoms	During Acute COVID-19 Infection	12 Months Post-COVID-19 Infection			
	$\bar{x}$ ± SE	$\bar{x}$ ± SE	Change $\bar{x}$ ± SE	*t*	*p*
**Loss of smell**	67.6 ± 42.3	15.2 ± 25.9	52.4 ± 41.4	12.788	**<0.001**
**Loss of taste**	59.7 ± 40.0	9.6 ± 20.7	50.2 ± 43.2	11.724	**<0.001**
**Nasal obstruction**	24.2 ± 32.4	8.0 ± 17.0	16.1 ± 31.6	5.149	**<0.001**
**Nasal discharge**	31.8 ± 32.7	11.1 ± 21.3	20.7 ± 36.2	5.788	**<0.001**
**Facial pain**	21.6 ± 33.6	4.1 ± 12.2	17.4 ± 34.2	5.146	**<0.001**
**Cough**	44.1 ± 6.5	6.4 ± 16.7	37.6 ± 36.5	10.412	**<0.001**
**Dyspnea**	27.1 ± 31.4	7.3 ± 16.4	19.9 ± 32.2	6.228	**<0.001**

**Table 3 life-12-01256-t003:** Change in olfactory and gustatory symptoms 12 months after infection in 102 positive COVID-19 patients.

Intensity of Symptoms during Acute COVID-19 Infection	Intensity of Symptoms 12 Months Post-COVID-19 Infection					
Loss of Smell	No	Mild	Moderate	Severe	Total	CI 95%
**No**	23	1	0	0	24 (23.5%)	[15.7, 33.0]
**Mild**	1	2	0	0	3 (2.9%)	[0.6, 8.4]
**Moderate**	6	4	1	0	11 (10.8%)	[5.5, 18.5]
**Severe**	29	20	7	8	64 (62.7%)	[52.6, 72.1]
**Total**	59 (57.8%)	27 (26.5%)	8 (7.8%)	8 (7.8%)	102 (100%)	-
**CI 95%**	[47.7, 67.6]	[18.2, 36.1]	[3.4, 14.9]	[3.4, 14.9]	-	-
**Loss of taste**	**No**	**Mild**	**Moderate**	**Severe**	**Total**	**CI 95%**
**No**	24	0	1	1	26 (25.5%)	[17.4, 35.1]
**Mild**	2	0	0	0	2 (2.0%)	[0.2, 6.9]
**Moderate**	12	8	1	1	22 (21.6%)	[14.0, 30.8]
**Severe**	33	11	6	2	52 (51.0%)	[40.9, 61.0]
**Total**	71 (69.6%)	19 (18.6%)	8 (7.8%)	4 (3.9%)	102 (100%)	-
**CI 95%**	[59.7, 78.3]	[11.6, 27.6]	[3.4, 14.9]	[1.1, 9.7]	-	-

No: (VAS = 0); Mild: (VAS > 0–30); Moderate: (VAS > 30–70); Severe: (VAS > 70–100) according to EPOS 2020 [19].

**Table 4 life-12-01256-t004:** Time of recovery (in weeks) of olfactory and gustatory dysfunction related to COVID-19.

	Loss of Smell			Loss of Taste		
Time for Recovery	*n*	%	95% CI	*n*	%	95% CI
≤2 weeks	18	50.0	[32.9, 67.1]	27	57.4	[42.2, 71.7]
2 ≤ 4 weeks	8	22.2	[10.1, 39.2]	9	19.1	[9.1, 33.3]
4 ≤ 24 weeks	10	27.8	[14.2, 45.2]	11	23.4	[12.3, 38.0]
24 < 48 weeks	0	0.0	[0, 0]	0	0.0	[0, 0]
**Total**	36			47		

**Table 5 life-12-01256-t005:** Crude and adjusted odds ratio (OR) and 95% CI for recovery of chemosensory dysfunction as a function of significant predictor variables.

Characteristics	Crude OR *^1^	95% CI	*p*	Adjusted OR *^2^	95% CI	*p*
**Age**						
≤40						
>40	0.20	[0.08, 0.52]	**0.010**	0.20	[0.07, 0.56]	**0.0023**
**Fever at the baseline**						
No						
Yes	3.19	[1.28, 7.94]	**0.0126**	2.75	[1.00, 7.61]	0.0511
**Duration of smell loss**						
≤4 weeks						
>4 weeks	0.25	[0.10, 0.64]	**0.0040**	0.27	[0.10, 0.76]	**0.0133**
**Duration of taste loss**						
≤4 weeks						
>4 weeks	0.31	[0.12, 0.80]	**0.0153**	-	-	-

*^1^ Estimated from binomial crude logistic regression. *^2^ Further adjusted by significant independent predictors (age, presence of fever at the baseline, and duration of smell and taste loss) according to stepwise multiple logistic regression model.

**Table 6 life-12-01256-t006:** Multiple linear regression modeling for change in VAS score at the baseline and 12 months after the COVID-19 infection.

Loss of Smell *^1^	Adjusted β ($\bar{x}$ ± SE)	95% CI	*p*	*R* ^2^
**Age**	−0.49 ± 0.22	[−0.93, −0.04]	**0.0327**	0.5446
**Duration of smell loss (weeks)**	−0.66 ± 0.23	[−1.10, −0.21]	**<0.0001**	
**VAS score for loss of smell at baseline**	0.86 ± 0.09	[0.68, 1.04]	**0.0045**	
**Loss of Taste *** ** ^2^ **	**Adjusted β ($\bar{x}$** **± SE)**	**95% CI**	** *p* **	** *R* ** ** ^2^ **
**Duration of taste loss (weeks)**	−0.79 ± 0.19	[−1.17, −0.42]	**<0.0001**	0.7496
**VAS score for loss of taste at baseline**	1.07 ± 0.07	[0.93, 1.21]	**<0.0001**	

*^1^ Model equation: Change in VAS score for loss of smell = 22.0 − 0.49 × (age) − 0.66 × (duration in weeks of smell loss) + 0.86 × (VAS score for loss of smell at the baseline). *^2^ Model equation: Change in VAS score for loss of taste = −10.5 − 0.79 × (duration in weeks of taste loss) + 1.07 × (VAS score for loss of taste at the baseline).

**Table 7 life-12-01256-t007:** Distribution of patients suffering parosmia 12 months after COVID-19 infection in relation to self-reported severity of olfactory loss both at the baseline and 12-month follow-up.

Intensity of Olfactory Loss at the Baseline (Acute COVID-19 Infection)	Intensity of Olfactory Loss 12 Months Post-COVID-19 Infection					
	No	Mild	Moderate	Severe	Total	CI 95%
**No**	1 *	0	0	0	1 (3.6%)	[0.1, 18.3]
**Mild**	0	1	0	0	1 (3.6%)	[0.1, 18.3]
**Moderate**	0	1	1	0	2 (7.1%)	[0.9, 23.5]
**Severe**	2	12	4	6	24 (85.7%)	[67.3, 96.0]
**Total**	3 (10.7%)	14 (50.0%)	5 (17.9%)	6 (21.4%)	28 (100%)	-
**CI 95%**	[2.3, 28.2]	[30.6, 69.4]	[6.1, 36.9]	[8.3, 41.0]	-	-

* This patient reported a qualitative smell disturbance (parosmia) with a normal perception of the odor intensity (VAS = 0).

**Table 8 life-12-01256-t008:** Crude and adjusted odds ratio (OR) and 95% CI for parosmia at the 12-month follow-up after the COVID-19 infection as a function of significant predictor variables.

Characteristics	Crude OR *^1^	95% CI	*p*	Adjusted OR *^2^	95% CI	*p*
**Loss of smell at baseline**						
≤30						
>30	6.63	[1.46, 30.23]	**0.0145**	6.27	[1.32, 29.87]	**0.0211**
**Loss of taste at baseline**						
≤30						
>30	7.04	[1.55, 32.0]	**0.0116**			
**Type of chemosensory dysfunction**						
Isolated						
Combined	5.37	[1.49, 19.42]	**0.0104**			
**Fever at baseline**						
No						
Yes	0.40	[0.16, 0.96]	**0.0407**			
**Duration of smell loss**						
≤12 weeks						
>12 weeks	4.53	[1.49, 13.78]	**0.0077**			
**Duration of taste loss**						
≤12 weeks						
>12 weeks	6.54	[1.96, 21.82]	**0.0023**	6.16	[1.74, 21.86]	**0.0049**

*^1^ Estimated from binomial crude logistic regression. *^2^ Further adjusted by significant independent predictors (moderate intensity of smell loss at the baseline and duration of taste loss) according to stepwise multiple logistic regression model.

**Table 9 life-12-01256-t009:** Distribution of olfactory UPSIT outcomes in relation to self-reported symptoms’ severity 12 months after COVID-19 infection.

Loss of Smell VAS 12-Months Post-COVID-19 Infection	UPSIT Outcome 12-Months Post-COVID-19 Infection						
	Normosmia	Mild Microsmia	Moderate Microsmia	Severe Microsmia	Anosmia	Total	95% CI
**No**	8	13	8	2	1	32 (46.4%)	[34.3, 58.8]
**Mild**	3	14	3	2	1	23 (33.3%)	[22.4, 45.7]
**Moderate**	1	2	2	1	1	7 (10.1%)	[4.2, 19.8]
**Severe**	0	1	2	2	2	7 (10.1%)	[4.2, 19.8]
**Total**	12 (17.4%)	30 (43.5%)	15 (21.7%)	7 (10.1%)	5 (7.2%)	69 (100%)	
**95% CI**	[9.3, 28.4]	[31.6, 56.0]	[12.7, 33.3]	[4.2, 19.8]	[2.4, 16.1]		

**Table 10 life-12-01256-t010:** Multiple linear regression modeling for UPSIT scores 12 months after the COVID-19 infection.

UPSIT Scores *^1^	Adjusted β ($\bar{x}$ ± SE)	95% CI	*p*	*R* ^2^
**Age**	−0.14 ± 0.05	[−0.23, −0.05]	**0.0042**	0.2294
**VAS score for loss of smell 12 months post-COVID-19 infection**	−0.07 ± 0.02	[−0.12, −0.03]	**0.0034**	

*^1^ Model equation: UPSIT score = 37.6 − 0.14 × (age) − 0.07 × (VAS score for loss of smell at 12 months).

## Data Availability

The data presented in this study are available on request from the corresponding author. The data are not publicly available due to ethical reasons.

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
