# Peer review of "Analysis of Prevalence and Predictive Factors of Long-Lasting Olfactory and Gustatory Dysfunction in COVID-19 Patients"

_life, 2022, doi:10.3390/life12081256_

Round 1

Reviewer 1 Report

An excellent aiming to assess the prognostic factors associated with increased risk of persistent chemosensory dysfunction in mild COVID-19 patients one year after the infection. The study is prospective, the statistics are appropriate, results are clearly presented , tables are relevant for the purpose of the study and discussions are comprehensive with heavily updated references. This is a must for every clinician involved in the care of COVID-19 patients.

Reviewer 2 Report

This is an interesting and important follow-up study on the chemosensory dysfunction due to COVID-19. It is well written, and I have only minor comments.

Table 1: There are rows in bold without any description on the reasons that they are in bold. Please either write the reasons or remove the ‘bold’ style.

Line 189-195: The sentence is broken in the middle. 

Line 204-205: Was there any correlation between chemosensory dysfunction and other symptoms? In line 218-219, there are some descriptions on correlation with other symptoms, and interestingly says a negative correlation between the fever at initial stage and reduced risk of chemosensory dysfunction after 12 months. This is very interesting.

Line 212-214: This is a very interesting, rare report on the development of chemosensory dysfunction later. Any specific symptoms or preexisting disease in these patients compared to others? As these are only 3 patients, may be good to insert a case report on them, if there are some specific symptoms or preexisting disease that could be related. Or, is it possible that they simply did not notice or report at the initial survey?

Reviewer 3 Report

This paper describes predictive factors of long-lasting olfactory and gustatory dysfunction in Covid-19 patients. Because olfactory and gustatory dysfunction has been reported as one of the primary long-covid syndromes, I read the paper with interest.

My questions and suggestions are as follows;

#1. Table 5

Line 215-216: The severity of symptoms at baseline was not associated with the persistence of chemosensory dysfunction at 12 months (Table 5)

When I read this for the first sentence in the paragraph, I could not find the ‘the severity of symptoms’ in Table 5. Afterward, I understood that only significant predictors were listed in Table 5 (footnote of Table 5).

It is easy for readers to understand when line 215-216 moves to the last part of the paragraph.

What does ‘the severity of symptoms’ mean?

Does the severity of symptoms include various symptoms? Or does it mean comparing each symptom's infection stage and one-year later stage?

Line 109-112: Self-reported scores quantifying the severity of symptoms during the COVID-19 infection; such as loss of smell, loss of taste, nasal obstruction, nasal discharge, facial pain, cough, and dyspnea, were reported using numerical visual analog scales (VAS) ranging from 0 to 100, according to guideline EPOS 2020 [18].

Because No.18 reference (Fokkens WJ et al.) has more than 400 pages, please explain the severity of symptoms concisely here.

#2. Table 6

PDF showed abnormal line changes in several places. Confirm a 95% confidence interval (CI).

The relationship between p-value and confidence interval does not seem compatible in several items in Table 6.

#3. Table 8

Fever (baseline) seems to be associated with no complaint of parosmia at the 12-month follow-up post-Covid-19 infection. Please explain fever (baseline) Yes. Is it fever at acute Covid-19 infection? If so, how was the temperature?

Many papers reported that young persons and females have a high rate of fever after the Covid-19 vaccination. Why was the crude odds ratio (OR) significant despite no significant adjusted OR?

#4. Table 9 and Fig. 2

The correlation between UPSIT scores and self-reported VAS for loss of smell 12 months after the COVID-19 infection was moderate. Was subjective smell sensation immediately after the COVID-19 infection associated with the UPSIT scores? I would like to know whether a relationship exists between the change of subjective sensation during the 12 months and the UPSIT scores.

#5. Discussion

Line 391-392: Conversely, having had fever during the acute infection resulted in acting as a protective factor for parosmia. I found the description based on Fever (baseline) in Table 8. Please explain more detailed conditions and the reason for the relationship.

Many studies reported a tendency for males to evaluate their subjective smell function better than females compared to the smell test results. Were there such findings in this investigation?
